# Physics-Guided Neural Forecasts of Nearshore Harmful Algal Blooms in the California Current System

## Abstract

Harmful algal blooms (HABs) are increasing in frequency, duration, and extent along the California coast, driven by climate variability, nutrient enrichment, and complex physical–biogeochemical interactions. Forecasting HAB development and spread remains a challenge, especially in Eastern Boundary Upwelling Systems where advection, stratification, and episodic river inputs strongly shape bloom dynamics. Existing approaches often trade physical realism for statistical flexibility, limiting generalization across bloom regimes. We present a physics-informed deep learning framework for nearshore chlorophyll-*a* forecasting in the California Current System, integrating multi-sensor satellite products, atmospheric and ocean reanalysis fields, and static geospatial predictors. Three architectures are evaluated: a convolutional long short-term memory network (ConvLSTM), a Temporal Fusion Transformer (TFT), and a physics-informed ConvLSTM (PINN) incorporating the two-dimensional advection–diffusion equation as a soft training constraint. A multi-year, 4 km-resolution dataset (2003–2021) is processed via a tailored feature engineering pipeline with quality-controlled gap-filling, rolling statistics, lagged predictors, and climatology-based anomalies. Models are assessed with strict spatiotemporal cross-validation, emphasizing spatial fidelity, bloom footprint representation, and predictor interpretability. Post-hoc explainability analyses identify key environmental drivers consistent with known upwelling–bloom linkages in the region. We present comparative skill assessments, spatial bias analyses, and predictor attribution results, highlighting the advantages and trade-offs of adding physical constraints to coastal HAB forecasting models. This work delivers a scalable and transferable methodology with direct implications for ecosystem management, fisheries, and public health.

## 1 Introduction

Harmful algal blooms (HABs) are ecological disturbances caused by rapid algal accumulation, often triggered by excess nutrients, warming waters, and stratification Litchman (2022); Ralston & Moore (2020). Their frequency and severity are increasing globally due to climate change and anthropogenic enrichment. Along the California coast, blooms of taxa such as *Karenia brevis*, *Pseudo-nitzschia*, and cyanobacteria pose major risks to ecosystems and human health Ryan et al. (2017); Thorel et al. (2017), leading to wildlife mortality, shellfish toxicity, respiratory illness, drinking-water contamination, and economic disruption to fisheries and tourism Grattan et al. (2016).

These stakes motivate early-warning systems that forecast bloom development and spread Moore et al. (2008), enabling managers to issue closures or adjust operations Molares-Ulloa et al. (2024). Yet current HAB prediction systems remain limited in coverage, generalizability, and mechanistic realism Marrone et al. (2023). We use chlorophyll-*a* as a bloom proxy given its accessibility from satellites and widespread use in surveillance Demiray et al. (2025).

Forecasting approaches fall into two categories. Process-based models capture ecological theory and ocean transport but require species-specific tuning, are sensitive to parameter uncertainty, and are computationally intensive Franks (1997); McGillicuddy (2010). Statistical and machine learning

(ML) models exploit empirical correlations across drivers Demiray et al. (2025), but risk overfitting, poor extrapolation, and physically implausible outputs Anderson et al. (2019); Park et al. (2024).

Recent work has explored hybrid models that couple ML with governing equations. Physics-informed neural networks (PINNs) embed constraints such as advection–diffusion directly into training, improving data efficiency and physical plausibility Raissi et al. (2019). Such approaches have been applied across geophysical domains including ocean circulation, hydrology, and solar irradiance.

Within HAB research, ML has been used primarily for time series or classification on satellite chlorophyll. Hill et al. developed *HABNet*, a convolutional–recurrent system for Florida blooms, achieving 8-day predictive skill but limited species and regional scope Hill et al. (2020). Molares Ulloa et al. compared advanced classifiers for harvest closure prediction across estuaries Molares-Ulloa et al. (2024). Cruz et al. reviewed ML applications for HABs, highlighting growing model complexity but persistent gaps in accounting for physics or generalizing across space Cruz et al. (2021). Gonzalez argued for trait-based ecological models to improve robustness under environmental change Litchman (2022). Despite progress, few HAB forecasting frameworks explicitly model fluid transport or spatial spread, both critical for actionable coastal predictions.

This work advances the intersection of ML and geophysics by testing whether physics-constrained neural networks improve chlorophyll-*a* forecasting along coastal California. Using a multi-year, multi-sensor dataset restricted to a ∼10-mile nearshore strip, we evaluate both pixelwise skill and bloom-scale spatial structure across chlorophyll regimes. Our hypothesis is that embedding advection–diffusion physics in a spatiotemporal network will improve 8-day forecast skill, particularly for high-chlorophyll events and under temporal–spatial distribution shifts. We further assess predictor attribution to identify physically meaningful drivers (e.g., wind stress curl, $Kd_{490}$, SST, river proximity) and their seasonal/regional variability. Overall, we aim to deliver a scalable, interpretable, and transferable framework for operational coastal bloom prediction.

## 2 METHODS

### 2.1 STUDY DOMAIN

We analyze a ∼10-mile (∼16 km) coastal strip along California, spanning the Southern California Bight to the Oregon spillover zone (Figure 1). This region includes ecosystems shaped by upwelling, river discharge, mesoscale eddies, and frontal dynamics Anderson et al. (2008). The domain is defined with a binary coastal mask applied to Level-3 satellite composites, retaining only ocean pixels within ≤16 km of shore (Natural Earth coastlines, EPSG:32610) and excluding land and offshore waters. The grid contains $240 \times 240$ cells at ∼4 km resolution, restricted to valid ocean pixels. The dataset spans 18.6 years (Jan 2003–mid-2021) with 851 non-overlapping 8-day steps ($\Delta t = 691{,}200$ s), capturing seasonal to decadal variability including ENSO, marine heatwaves, and upwelling shifts Jacox et al. (2016). Submesoscale filaments (<2 km) remain unresolved Le Traon et al. (2015), but smaller-scale variability is partly represented by predictors such as distance-to-river, wind stress curl, and sea surface height gradients. The mask encompasses key ecological and management zones, including Monterey Bay National Marine Sanctuary, major estuaries (e.g., Eel, Navarro), and the Southern California Bight. This consistent record, paired with reanalysis drivers, underpins our physics-informed machine learning experiments on coastal HAB dynamics.

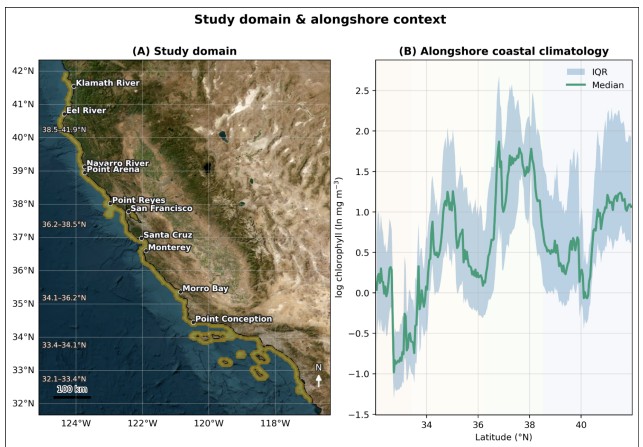

Figure 1: Study domain and coastal mask: $\sim$10-mile nearshore strip, grid resolution ($\sim$4 km), and key regions (SoCal Bight, Monterey Bay, Northern CA).

## 2.2 DATA SOURCES AND TRANSFORMATION

Table 1 summarizes predictors, sources, and relevance. Variables capture processes affecting phytoplankton, including nutrient supply, light, mixing, and circulation.

Ocean colour products are from MODIS-Aqua Level-3 8-day composites at $\sim$4 km resolution, reducing cloud effects while retaining bloom dynamics. Chlorophyll-$a$ is log-transformed with floor $\varepsilon = 0.056616$ mg m$^{-3}$ and a binary floor flag to reduce variance and heavy tails. Atmospheric drivers are from ERA5 reanalysis, aligned with MODIS windows. Wind stress is computed with the quadratic drag law,

$$\boldsymbol{\tau} = \rho_a\, C_D\, \|\mathbf{u}_{10}\| \cdot \mathbf{u}_{10}, \tag{1}$$

and wind stress curl is derived by centered differences with $\cos(\varphi)$ scaling Large & Pond (1981).

Ocean state variables come from CMEMS GLORYS reanalysis. Derived fields include current speed, divergence, vorticity, and SSH gradient magnitude,

$$|\nabla\eta| = \sqrt{\left(\frac{\partial\eta}{\partial x}\right)^2 + \left(\frac{\partial\eta}{\partial y}\right)^2}, \tag{2}$$

capturing mesoscale fronts and geostrophic shear Morrow et al. (2004). Additional features include rolling statistics (24–40 d), monthly anomalies relative to a training climatology, and 1–3 timestep lags. River influence is parameterized by distance-to-river and exponential decay $\exp(-d/\ell)$, alongside static predictors (latitude, coast distance, river rank). All variables are quality-controlled, collocated to the mask, standardized with training-period statistics, and stored as aligned spatiotemporal arrays.

## 2.3 FEATURE ENGINEERING

All predictors were mapped to the $\sim$4 km coastal grid, quality-controlled, and standardized with training-period statistics. Ocean-colour variables (chlorophyll-$a$, Kd$_{490}$, nFLH) were gap-filled to address cloud contamination using spatiotemporal interpolation biased along prevailing winds to preserve anisotropic transport typical of upwelling systems. Chlorophyll-$a$ and Kd$_{490}$ were processed in log space with median–seasonal baselines and caps, while nFLH was imputed in linear space with IQR constraints. A supplementary climatology routine bridged longer gaps and clipped synthetic values to the observed 1st–99th percentile. From dynamic drivers we derived rolling statistics (24–40 d), monthly anomalies, and 1–3 timestep lags; static features included latitude, distance to coast and rivers, and an exponential river influence index. Spatial derivatives (e.g., wind stress curl, SSH gradients, divergence, vorticity) were computed in projected coordinates with $\cos(\varphi)$ scaling. To counter the natural skew toward low chlorophyll, training was stratified into four linear-space regimes (Background–Extreme), with balanced sampling and class-weighted losses ensuring adequate representation of bloom events.

Table 1: Predictor variables, data sources, and relevance for HAB forecasting

| Predictor (units) | Data source | Relevance |
|---|---|---|
| Chlorophyll-a (mg m$^{-3}$) | NASA Earthdata | Primary biomass proxy; basis for bloom detection and chl-derived features. |
| nFLH (mW cm$^{-2}$ $\mu$m$^{-1}$ sr$^{-1}$) | NASA Earthdata | Proxy for chlorophyll fluorescence; flags high-biomass events. |
| Kd$_{490}$ (m$^{-1}$) | NASA Earthdata | Water clarity; constrains euphotic depth and light availability. |
| 10 m wind (u) (m s$^{-1}$) | ERA5 | Zonal wind; alongshore component driving upwelling/downwelling. |
| 10 m wind (v) (m s$^{-1}$) | ERA5 | Meridional wind; cross-shore transport and mixing. |
| Wind speed (m s$^{-1}$) | ERA5 | Turbulent mixing; modulates nutrient and light supply. |
| Wind stress $|\tau|$ (N m$^{-2}$) | ERA5 | Ekman transport; indicator of upwelling-favorable periods. |
| Shortwave radiation (W m$^{-2}$) | ERA5 | Photosynthetically active radiation; interacts with stratification. |
| Precipitation (mm day$^{-1}$) | ERA5 | Freshwater pulses; potential nutrient and turbidity effects nearshore. |
| 2 m temperature (K) | ERA5 | Atmosphere–ocean heat flux; influences SST and stratification. |
| 2 m dewpoint (K) | ERA5 | Humidity and latent heat flux; linked to mixing and clouds. |
| Eastward current, uo (m s$^{-1}$) | CMEMS | Alongshore advection of biomass and nutrients. |
| Northward current, vo (m s$^{-1}$) | CMEMS | Cross-frontal transport and retention of blooms. |
| Current speed (m s$^{-1}$) | CMEMS | Net transport intensity; dispersion/retention of biomass. |
| Divergence (s$^{-1}$) | CMEMS | Convergence/divergence zones affecting nutrient supply and retention. |
| Vorticity (s$^{-1}$) | CMEMS | Eddies and fronts; mesoscale trapping of blooms. |
| SSH (m) | CMEMS | Large-scale circulation and mesoscale features. |
| SSH gradient magnitude (m m$^{-1}$) | CMEMS | Frontal intensity; proxy for geostrophic shear. |
| SSS (PSU) | CMEMS | Freshwater influence from rivers; buoyancy control. |
| SST (°C) | CMEMS | Thermal control of growth rates and stratification. |

## 2.4 CROSS-VALIDATION

We used blocked, stratified spatiotemporal cross-validation to test generalization under distribution shifts while balancing chlorophyll regimes. Three modes were evaluated: (A) temporal holdout (2003–2015 train, 2016–2018 validation, 2019–mid-2021 test), (B) spatial holdout (one of $N_s = 5$ alongshore bands withheld), and (C) combined temporal+spatial (*STRICT_OOS*). Metrics were computed in both log- and linear-space chlorophyll, with sampling stratified by month-adjusted regimes (Background–Extreme). For an operational-style forecast, models were retrained on 2003–2018 and evaluated on 2019–mid-2021, simulating real-world prediction of future blooms along a known coastline.

## 2.5 MODEL ARCHITECTURE

We compare three approaches for forecasting nearshore chlorophyll-*a*: a convolutional LSTM (ConvLSTM), a Temporal Fusion Transformer (TFT), and a physics-informed ConvLSTM (PINN). All models take $L_{\text{in}}$ historical 8-day frames of predictors and output chlorophyll-*a* at $t + \Delta t$ ($\Delta t = 8$ days) Reichstein et al. (2019).

ConvLSTM integrates convolutional layers into an LSTM to capture spatial and temporal dependencies. While effective at advective features, it often smooths gradients and underpredicts extremes Amato et al. (2020).

TFT adapts an attention-based sequence model for coastal grids by flattening valid points and appending static metadata (latitude, coast distance, river distance). Variable selection, gating, and attention enable heterogeneous drivers and long-range dependencies. Training uses a quantile regression loss ($\tau = \{0.1, 0.5, 0.9\}$), with $\tau = 0.5$ for point forecasts and full quantiles for probabilistic metrics Lim et al. (2021); Pathak et al. (2022).

PINN augments the ConvLSTM with a residual loss enforcing the 2-D advection–diffusion equation in standardized log-chlorophyll space with a learnable source $S(x, y, t)$:

$$\frac{C^{t+\Delta t} - C^t}{\Delta t} + u\frac{\partial C}{\partial x} + v\frac{\partial C}{\partial y} - \kappa\nabla^2 C - S = 0 \tag{3}$$

where $C$ is log chlorophyll, $\mathbf{u} = (u, v)$ is reanalysis surface velocity, and $\kappa$ is a non-negative diffusivity (softplus-initialized at 25 m$^2$ s$^{-1}$) Marchesiello et al. (2003). Derivatives use convolutional stencils with metric scaling; Neumann boundaries are applied coastward, radiation boundaries offshore Marchesiello et al. (2001).

Training proceeds in two phases: a class-weighted Huber loss on $\log C$ with stratified mini-batches across chlorophyll regimes, followed by introduction of the physics residual with a ramped weight $\lambda \in [0, 1]$ once validation RMSE plateaus Raissi et al. (2019).

## 2.6 EVALUATION METRICS

We assessed skill with (i) continuous metrics in log/linear chlorophyll ( RMSE, MAE Willmott & Matsuura (2005), KGE Gupta et al. (2009), Spearman's $\rho$, Pearson's $r$), (ii) categorical bloom detection at $5 \, \mathrm{mg} \, \mathrm{m}^{-3}$ (probability of detection, false alarms, precision/recall, $F_1$, ROC-AUC, IoU, centroid displacement Sokolova & Lapalme (2009)), and (iii) spatial fidelity (normalized RMSE, relative scale error, spectral energy ratios Arbic et al. (2012)). Probabilistic TFT forecasts were additionally evaluated with CRPS Hersbach (2000), while PINN diagnostics included physics-residual distributions Raissi et al. (2019).

## 2.7 PREDICTOR IMPORTANCE ANALYSIS

We interpret model forecasts using post hoc explainability methods for both baselines and deep networks. For pixelwise GBDT baselines and aggregated ConvLSTM/TFT inputs, we compute Shapley additive values (SHAP) to attribute predictor contributions at each timestep, with global importance obtained by averaging absolute values across all valid pixels Lundberg & Lee (2017).

For ConvLSTM and PINN, SHAP (via a GBDT surrogate) is complemented with direct *occlusion* and *permutation* importance on input channels, verifying that surrogate attributions align with network sensitivities.

To ensure physical interpretability, we use partial dependence plots (PDPs) and bivariate scatterplots for top predictors (e.g., wind stress curl, $Kd_{490}$, SST) Friedman (2001). PDPs reveal nonlinear thresholds and saturation effects, while scatterplots validate attribution-derived relationships against empirical data.

Together, these methods combine model-agnostic SHAP for comparability with perturbation tests for deep models, framing predictor influence in a physically consistent context.

## 2.8 CASE STUDY EVENT ANALYSIS

To complement domain–wide evaluation, we examine two notable HAB episodes in the test period (2019–2021): (i) the Navarro River lagoon bloom (39.19°N, 124.00°W; 24 July 2020) and (ii) the Monterey Bay event near the Coast Guard dock (36.61°N, 121.89°W; 25 May 2021). For each, the relevant 8-day composite was extracted from a fixed coastal subset, with Monterey used as the illustrative example. Coastal fidelity was maintained by restricting plots to ocean pixels within 10 miles of the shoreline using a UTM-projected land mask. Fields were interpolated and lightly smoothed within the mask to improve readability, while land and offshore areas were left blank. Panels were rendered with consistent contrast scaling (gamma = 0.85). These case studies provide event-scale validation of model performance under contrasting conditions: a broad, shelf-scale bloom in Monterey Bay and a smaller river-influenced bloom at Navarro.

## 2.9 USE OF ARTIFICIAL INTELLIGENCE TOOLS

ChatGPT (OpenAI, GPT-5, 2025) was used only for language editing and grammar refinement. It was not employed to generate scientific content, data, or analysis. All text was reviewed and verified by the authors for accuracy and compliance with ICLR ethical guidelines.

## 3 RESULTS

### 3.1 MODEL SKILL OVERVIEW

On the test set, ConvLSTM and PINN achieved similar performance, with the PINN showing small but consistent gains (Table 2). In log space, RMSE/MAE were 0.801/0.593 for ConvLSTM and 0.799/0.592 for PINN; correlations and KGE improved slightly ($\Delta \sim 0.004$). These results suggest the physics term mainly acted as a regularizer, improving generalization without changing bulk accuracy Raissi et al. (2019); Karniadakis et al. (2021).

The TFT performed slightly worse (RMSE 0.814, MAE 0.605), reflecting challenges of adapting transformer architectures to sparse coastal grids. The PINN also converged faster in training, while

all models showed negligible bias ($< 0.001$), consistent with prior HAB forecasting ceilings near RMSE $\sim 0.80$ Ralston & Moore (2020); Hill et al. (2020).

## 3.2 SPATIAL AND BIAS PATTERNS

Domain-averaged bias was negligible ($< 0.001$), but spatial diagnostics showed regional errors (Figure 2). Scatterplots clustered near the 1:1 line, consistent with high correlations, but with vertical spread from localized under- and overprediction.

Bias maps revealed underprediction by ConvLSTM and PINN in the Southern California Bight (e.g., Catalina, Los Angeles, San Diego), where optical complexity challenges all models Kim et al. (2009). The TFT instead overpredicted in high-latitude sectors near Oregon, reflecting difficulty with colder, nutrient-rich waters. ConvLSTM and PINN handled these gradients better, though some overestimation remained Marchesiello et al. (2003).

The PINN offered only modest improvement over the ConvLSTM, mainly damping extremes rather than altering large-scale bias patterns.

## 3.3 STRATIFIED PERFORMANCE AND PERSISTENCE BENCHMARKING

Forecast accuracy varied with bloom intensity (Figure 3A). The PINN had the lowest RMSE in three of four quartiles, with strongest gains in the mid-to-high range (Q2–Q3); errors were highest in rare extreme events ($>$Q3).

Against a persistence forecast (Figure 3B–C), the PINN outperformed across nearly all latitudes, with largest advantages in southern (Catalina–San Diego) and northern (California–Oregon) sectors. Monthly RMSE showed seasonal variability: lowest in fall–winter, highest in summer, when mesoscale activity degrades skill Frolov et al. (2013). The PINN consistently reduced RMSE relative to persistence, though gaps narrowed in summer.

Overall, the physics-informed approach offered greatest benefits in moderate-to-high bloom regimes, while persistence remained competitive in low-variability conditions.

## 3.4 ROBUSTNESS AND PHYSICAL CONSISTENCY DIAGNOSTICS

We evaluated spatial fidelity and physical realism with four diagnostics: relative scale error, spectral energy ratio, cross-shelf RMSE, and physics residuals (Figure 4). Across all, the PINN performed strongest, supporting our hypothesis that physics constraints improve spatial robustness.

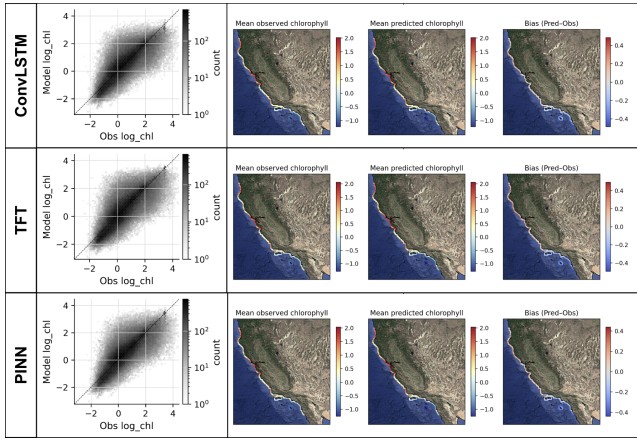

Figure 2: Domainwide performance and bias structure. Left: observed vs. predicted log chlorophyll scatter (density shading). Right: spatial bias maps for ConvLSTM, TFT, and PINN during the test period.

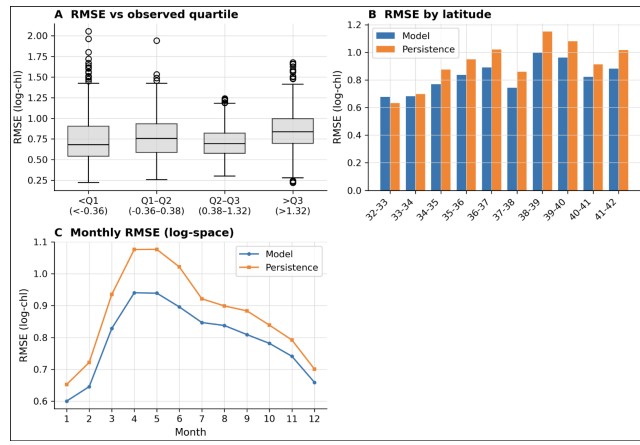

Figure 3: Stratified skill and persistence comparison. (A) RMSE by observed log-chlorophyll quartile; (B) model vs. persistence by latitude band; (C) monthly RMSE (log space).

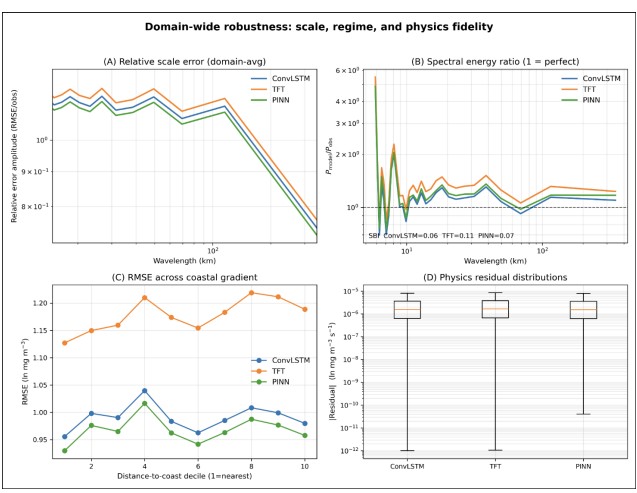

Figure 4: Spatial fidelity and physical consistency: (A) relative scale error; (B) spectral energy ratio across wavenumbers; (C) RMSE vs. distance-to-coast deciles; (D) physics residual distributions.

Relative scale error (Fig. 4A) showed the PINN best matched bloom extent and fragmentation, while the TFT over-smoothed or overextended features Lim et al. (2021). Spectral ratios (Fig. 4B) confirmed this, with ConvLSTM and PINN deviations ∼0.06–0.07 from unity versus 0.11 for the TFT. RMSE by distance-to-coast (Fig. 4C) revealed performance degradation offshore but consistently lower error for the PINN across all deciles. Physics residuals (Fig. 4D) were most tightly centered for the PINN, indicating closer compliance with the advection–diffusion constraint.

Together, these diagnostics show that the PINN produces more physically consistent and structurally realistic forecasts than either ConvLSTM or TFT.

## 3.5 PREDICTOR IMPORTANCE AND PHYSICAL INTERPRETATION

SHAP analysis of the PINN (Figure 5) identified five dominant predictors: diffuse attenuation at 490 nm ($Kd_{490}$), river distance and influence, sea surface temperature (SST), and shortwave radiation. These capture optical, hydrological, and physical drivers consistent with known bloom mechanisms.

$Kd_{490}$ highlighted light availability as a key control, strongest in early bloom states. River distance and influence emphasized freshwater plumes as episodic nutrient sources, while SST and shortwave

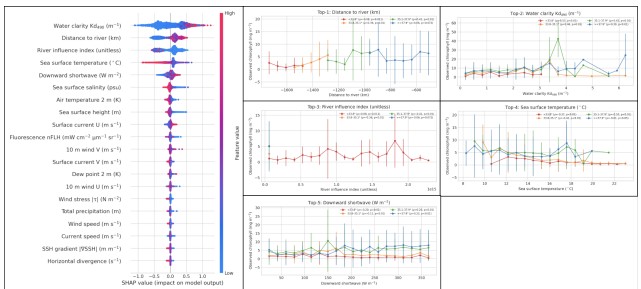

Figure 5: Predictor importance and response structure for PINN: SHAP global rankings and representative partial dependence curves (top predictors: $Kd_{490}$, river distance/influence, SST, shortwave).

reflected stratification–energy interactions. Correlations shifted across regimes: SST was negatively linked to low/moderate blooms but insignificant at extremes; shortwave flipped from weakly negative to positive across regimes. Partial dependence plots confirmed nonlinear thresholds (e.g., SST, $Kd_{490}$). Overall, the PINN recovered ecologically plausible drivers whose influence varied with bloom state, underscoring its interpretability for HAB forecasting.

### 3.6 CASE STUDIES

We examined two contrasting test events: a broad HAB in Monterey Bay (25 May 2021) and a localized river-influenced HAB at Navarro Lagoon (24 July 2020) (Figure 6).

In Monterey Bay, all models reproduced the bloom core, with ConvLSTM and PINN better resolving offshore gradients and reducing southern overprediction compared to TFT. At Navarro, both convolutional–recurrent models localized the bloom near the river mouth, while TFT overpredicted coastal spread—consistent with its higher scale error (Section 4.4).

Overall, the models captured large-scale bloom morphology and drivers but missed fine-scale features unresolved at 4 km resolution, underscoring the need for higher-resolution inputs and dynamic river data for improved fidelity.

### 3.7 LEAD-TIME DEPENDENCE

We evaluated multi-lead forecasts at 8–24 day horizons (Table 3). Performance declined monotonically across models, reflecting growing driver uncertainty and coastal variability. ConvLSTM and PINN showed nearly identical trends, with the TFT consistently weaker.

The PINN's advantage was most evident at short leads (8–16 d), where the physics constraint improved stability and fidelity. At longer horizons, skill converged as unresolved mesoscale dynamics and missing driver updates limited all models.

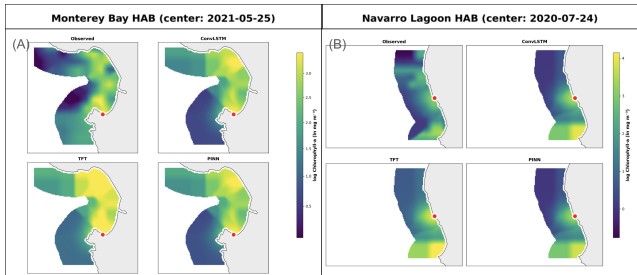

Figure 6: Case studies. Left: Monterey Bay HAB (25 May 2021); Right: Navarro Lagoon HAB (24 July 2020). Observed vs. forecast chlorophyll and bloom footprint.

Table 2: Lead-time performance metrics for ConvLSTM, TFT, and PINN models. Skill is relative to persistence baseline, in both log and linear chlorophyll-$a$ space.

| Model & Lead | RMSE (ln chl) | Skill (ln chl) [%] | RMSE (mg m$^{-3}$) | Skill (mg m$^{-3}$) [%] | $r$ (ln chl) |
|---|---|---|---|---|---|
| **ConvLSTM** | | | | | |
| 8 d | 0.801 | 9.571 | 5.106 | 17.627 | 0.773 |
| 16 d | 0.909 | 10.953 | 5.517 | 19.546 | 0.708 |
| 24 d | 0.946 | 10.661 | 5.668 | 19.689 | 0.684 |
| 32 d | 0.973 | 10.161 | 5.760 | 19.440 | 0.666 |
| **TFT** | | | | | |
| 8 d | 0.814 | 8.102 | 5.107 | 17.608 | 0.777 |
| 16 d | 0.926 | 9.257 | 5.594 | 18.418 | 0.712 |
| 24 d | 0.965 | 8.892 | 5.781 | 18.089 | 0.687 |
| 32 d | 0.992 | 8.343 | 5.886 | 17.686 | 0.669 |
| **PINN** | | | | | |
| 8 d | 0.799 | 9.725 | 5.123 | 17.343 | 0.777 |
| 16 d | 0.908 | 11.002 | 5.579 | 18.641 | 0.712 |
| 24 d | 0.946 | 10.640 | 5.752 | 18.509 | 0.687 |
| 32 d | 0.973 | 10.125 | 5.851 | 18.164 | 0.669 |

Overall, the PINN offered modest but consistent gains, especially in structural fidelity at short leads, supporting physics-informed networks as a useful—though not sufficient—advance for operational HAB forecasting.

## 4 DISCUSSION

We tested whether embedding the advection–diffusion equation into a deep spatiotemporal network improves HAB forecasting along coastal California. Results partially support this hypothesis: the PINN matched or modestly outperformed the ConvLSTM in bulk metrics, with clearer gains in spatial fidelity and at short (8–16 day) leads.

These findings align with prior physics–ML work showing that constraints guide models toward physically plausible solutions Raissi et al. (2019); Reichstein et al. (2019). The PINN better preserved bloom morphology across the shelf and reduced relative scale error and spectral distortions, consistent with the role of horizontal advection in shaping bloom dynamics Gruber et al. (2011). Improvements were modest in RMSE and correlation ($\Delta \sim 0.002$–$0.004$), reflecting a skill ceiling set by coarse inputs (4 km, 8-day composites) that cannot resolve submesoscale features or rapid bloom events Le Traon et al. (2015). Similar limits have been noted in other operational HAB forecasts Frolov et al. (2013); Anderson et al. (2019).

Skill declined for extreme chlorophyll and at longer leads, echoing known predictability horizons of 2–3 composite periods Jacox et al. (2016). Deterministic forecasts beyond this range likely require driver-forecast coupling or ensemble methods. Attribution analyses identified $Kd_{490}$, river proximity, SST, and shortwave radiation as dominant predictors—drivers consistent with upwelling–nutrient–light controls in the region Anderson et al. (2008); Horner et al. (1997). Their emergence without species-level tuning underscores the ecological interpretability of physics-guided ML.

Operationally, the PINN's reduced false spread and improved bloom coherence translate into fewer false alarms and more targeted advisories, though persistent high-latitude biases highlight missing processes such as plume–current interactions.

Future work should integrate higher-resolution ocean color and dynamic reanalysis forecasts, explore multi-task predictions (e.g., chlorophyll+nitrate), and test transferability to other upwelling systems. While not transformative in bulk skill, physics-informed networks provide consistent, interpretable, and spatially realistic improvements, an important step toward reliable, actionable HAB forecasting.

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
