# OpenReview forum: "Physics-Guided Neural Forecasts of Nearshore Harmful Algal Blooms in the California Current System"
_ICLR.cc/2026/Conference — Submitted to ICLR 2026_

### Official Review · Reviewer_CSUo · 2025-10-28

**Soundness:** 3
**Presentation:** 3
**Contribution:** 2
**Rating:** 2
**Confidence:** 4

**Summary:**

This paper proposes a physics-informed deep learning framework for forecasting nearshore harmful algal blooms (HABs) along the California coast. Using multi-sensor satellite data, ocean–atmosphere reanalysis fields, and geospatial predictors, the authors compare three models: a ConvLSTM, a Temporal Fusion Transformer (TFT), and a physics-informed ConvLSTM (PINN) that embeds a 2-D advection–diffusion equation as a soft constraint.
The model trained on an 18-year, 4-km dataset with blocked spatiotemporal cross-validation, the PINN achieves modest but consistent gains in spatial fidelity and generalization, particularly for moderate-to-high bloom regimes and short lead times (8–16 days). SHAP and partial-dependence analyses reveal interpretable physical drivers (e.g., Kd490, SST, river proximity), enhancing ecological insight. Overall, the work offers a scalable and interpretable approach for physics-guided coastal forecasting with direct implications for environmental management.

**Strengths:**

S1: The paper has Figures that are well-annotated, equations properly derived, and every methodological step, feature engineering, model architecture, evaluation, and explainability are presented in reproducible detail. The explicit disclosure of limited LLM use for grammar use only shows ethical transparency.

S2: The paper shows prediction accuracy by representing physical interpretability through SHAP and partial-dependence analyses. Showed key environmental drivers such as Kd490, SST, and river influence improve scientific understanding and credibility, thus making an important advance in a field often dominated by black-box models.

S3: The author's contribution is by bringing physics-informed neural networks (PINNs) into the domain of nearshore harmful algal bloom (HAB) forecasting, a field traditionally dominated by empirical statistical models or purely mechanistic simulations.

**Weaknesses:**

W1: The paper ends with a detailed discussion, but no explicit conclusion. A dedicated “Conclusion” section is missing, making it difficult to understand the what are final takeaway.

W2: Generality beyond the California coast is suggested but not very clearly demonstrated.

W3: The probabilistic skill is assessed only for TFT, but PINN uncertainty quantification is absent. The PINN evaluation in Section 2.6 only uses basic error metrics and briefly mentions physics residuals without defining how they’re measured or interpreted.

W4: The paper does not have a Related Work section explaining how the related studies have been conducted and how they are different from the proposed approach. There have been many PINN approaches for different applications. They should be clearly compared.

**Questions:**

Q1: In Section 2.5, the author introduces the residual-loss weight (λ) and the diffusivity parameter (κ = 25 m² s⁻¹) as key components of the physics constraint. Could you clarify whether you tested different values or schedules for these parameters? If not, how are you sure that the reported improvements are robust to changes in λ and κ, rather than dependent on a specific setting?

Q2: It is mentioned in the Abstract and Discussion that the framework is scalable and transferable to other upwelling systems. Could you elaborate on whether this transferability is theoretical or based on preliminary cross-region experiments? If not tested yet, what modifications would be needed to adapt the model to other coastal systems?

Q2: In Section 3.4, the author evaluates spatial fidelity using metrics such as relative scale error, spectral energy ratio, and cross-shelf RMSE, concluding that the PINN improves spatial robustness. Could you clarify how you define “high spatial fidelity” and how fidelity differs from accuracy? What aspects of the physics constraint most contribute to this improvement?

---

### Official Review · Reviewer_amy1 · 2025-10-29

**Soundness:** 2
**Presentation:** 1
**Contribution:** 2
**Rating:** 2
**Confidence:** 4

**Summary:**

This paper addresses the challenge of forecasting nearshore harmful algal blooms (HABs) in the California Current System. The tackle this, the paper propose a physics-informed deep learning framework that integrates multi-source data, including satellite products and reanalysis fields. They evaluate three architectures: a ConvLSTM, a Temporal Fusion Transformer (TFT), and a physics-informed ConvLSTM (PINN) that incorporates the 2D advection-diffusion equation as a soft constraint. Trained on an 18-year dataset, the physics-informed model offers marginal quantitative gains but more spatially coherent forecasts than a standard ConvLSTM. Its predictions also align with known physical drivers like upwelling.

**Strengths:**

1. Addresses the real-world challenge of HAB forecasting and applies an interpretable physics-informed model (PINN).
2. Features comprehensive data preparation and experimental analysis.

**Weaknesses:**

1. The work primarily applies existing models to a new domain with little methodological innovation.

2. The reported improvements are too small to be practically meaningful.

3. Lacks key details on model architecture, hyperparameters, and experimental setup.

4. Lack of comparison against more baselines, such as the transformer models.

5. Figures and tables are often unclear and difficult to read.

**Questions:**

1. Could you provide the specific hyperparameters and architectural details for your models?

2. Why were transformer-based forecasting models not included as a baseline for comparison?

3. Can the framework handle multivariate coupling (e.g., nitrate, temperature) to model ecosystem processes beyond chlorophyll-a?

4. How does the computational cost of training the PINN compare to the ConvLSTM and TFT models?

5. Regarding the limited novelty, could you clarify what you see as the core methodological innovation of this work beyond applying existing techniques to a new dataset?

---

### Official Review · Reviewer_bfTY · 2025-11-01

**Soundness:** 2
**Presentation:** 2
**Contribution:** 2
**Rating:** 2
**Confidence:** 3

**Summary:**

The paper proposes physics-guided neural forecasts of nearshore harmful algal blooms (HABs) along the California coast. It assembles an 18.6-year, ~4 km dataset of satellite ocean color, atmospheric ERA5 fields, ocean reanalyses, and static geospatial predictors, restricted to a ~10-mile coastal strip. Three models are compared for 8–32-day chlorophyll-a forecasting: ConvLSTM, Temporal Fusion Transformer, and a physics-informed ConvLSTM trained with an advection–diffusion residual as a soft constraint. The PINN yields modest but consistent improvements over ConvLSTM in correlation and notably better spatial fidelity/scale error, especially at 8–16 days. Gains are small in bulk RMSE.

**Strengths:**

1. The paper presents a well-defined and societally relevant forecasting task focused on coastal harmful algal blooms.
2. The paper proposes a long and carefully curated dataset that integrates ocean, atmosphere, and geospatial variables. The dataset covers nearly two decades and is processed with clear feature engineering steps and masking choices that align with management applications.

**Weaknesses:**

1. The improvement over the baseline ConvLSTM is relatively small in terms of overall error metrics. The benefits of the physics constraint are primarily evident in spatial fidelity rather than in bulk RMSE or correlation, and the paper does not provide statistical tests to confirm whether these differences are statistically significant.
2. The physical constraint itself is limited to a simple linear advection–diffusion term with an unspecified source component. It does not incorporate biological growth, decay, or nutrient interactions, which limits how “physics-guided” the approach truly is.
3. The coarse spatiotemporal resolution of the inputs and targets is acknowledged as a limitation. Yet, the study does not attempt ablation experiments to verify that finer resolution or sub-weekly sampling would meaningfully increase skill.
4. The transformer baseline appears disadvantaged because it treats spatial points independently rather than leveraging spatial attention. Using more recent spatial-transformer designs could yield stronger baselines and make the comparison fairer.
5. The analysis of extreme events remains mostly qualitative. A more quantitative assessment would better support the claims.
6. Variability across training seeds or folds is not reported, and the details of model tuning, physics-loss scheduling, and reproducibility are only partially specified.

**Questions:**

1. Why apply the advection–diffusion constraint in log-chlorophyll space, and how sensitive are results to this choice?
2. Has the method been tested or considered for operational coastal bloom forecasting?
3. How sensitive are results to boundary condition settings and diffusivity initialization?
4. Does fixing the physics-loss weight instead of ramping it affect stability or performance?

---

### Official Review · Reviewer_CrZH · 2025-11-01

**Soundness:** 3
**Presentation:** 2
**Contribution:** 2
**Rating:** 4
**Confidence:** 3

**Summary:**

This paper assesses whether physics-informed constraints improve near-shore chlorophyll-a (as a HAB proxy) forecasting in the California Current. It compares a ConvLSTM, a TFT, and a physics-guided ConvLSTM with a soft advection–diffusion loss, trained on ~4-km, 8-day composites with multiple dynamic/static drivers.

**Strengths:**

- Clear hypothesis and goal of adding a physics loss to help forecast coastal chlorophyll/HABs.

- Physics guidance seems to help the most at near-term horizons and in spatial structure

**Weaknesses:**

- Performance of all models look the same and hardly enough gains quantitatively.

- Would be good to report uncertainty or statistical measure to further compare.

- Although the authors have provided qualitative analysis but it hardly compares all the baselines or shows sufficient evidence to understand why and how one is better.

- The paper is written in a sort of crytic manner from teh perspective of an ML reader, assuming much domain knowledge. It would be good to have an appendix section detailing all the introductory domain information, explaining datasets, ecological features and domain terms in detail.

- The paper has citations missing for the datasets mentioned. Such as ERA5/MODIS/CMEMS.

- In Fig. 2 the three models appear nearly indistinguishable; please quantify significance and show uncertainty bars.

- 8-day target is under-motivated. Why 8? Is it inherited from MODIS composites or operational cadence? Would be good to compare with 7 or 10? How does that impact model performance?

- The paper has minimal baselines for comparison. Fourier Neural Operator / AFNO have been shown to work on ocean and climate data, why not use those to compare as baselines? Or use HABnet?

- I am unsure of the novelty of this work, please elaborate on that. Right now it seems an applied work to me. While novelty is and should not be the main criteria, however this paper also lack insights of value. This work can benefit with more experiments and deeper discussions between various methods.

- The “binary floor flag” is mentioned but how/where it’s used isn’t obvious?

- Can you show more ablations such as: (i) raw vs monthly-anomaly features, (ii) dynamic vs static drivers, (iii) different λ schedules. Analyzing which ecological features impact the most or which are not of much use?

Minor

- Move domain background (drivers, sensors, acronyms) to a succinct “Data & Drivers” subsection with 3–4 line blurbs + references; explicitly cite ERA5, MODIS, GLORYS/CMEMS.

- Put full architecture and training hyperparams in Appendix: layer specs, kernels, stencils, normalization, masking, loss weights, schedulers, early stopping.

- Fig. 2 left should be color (grayscale makes curves indistinguishable), right panel is blurry even when zoomed.

- Include the structure-diagnostic plots (your Figs. 3–4 type) for all three models, not just the PINN, so it is clear if physics helps/hurts.

- In multi-column examples, clearly label which column is GT.

**Questions:**

- 8-day target is under-motivated. Why 8? Is it inherited from MODIS composites or operational cadence? Would be good to compare with 7 or 10? How does that impact model performance?
- The “binary floor flag” is mentioned but how/where it’s used isn’t obvious?
- The paper has minimal baselines for comparison. Fourier Neural Operator / AFNO have been shown to work on ocean and climate data, why not use those to compare as baselines? Or use HABnet?

---

### Meta-Review · Area_Chair_Yceo · 2026-01-04

**Summary:**

This paper compares several ML solutions for monitoring Harmful Algal Blooms in California. The paper is a direct comparison of methods that might be more suitable for an oceanography or environmental publication. For an applied paper of this nature to be accepted at ICLR would require a significant methodological contribution on the algorithmic side that has further implications than the proposed environmental problem.

**Reviewer Concerns:**

The paper is not an ML paper.

**Reviewer Scores:**

They would not change, as the authors did not engage.

---

### Decision · Program_Chairs · 2026-01-26

Reject